# Tomato Seed Coat Permeability: Optimal Seed Treatment Chemical Properties for Targeting the Embryo with Implications for Internal Seed-Borne Pathogen Control

Hilary Mayton [1,†], Masoume Amirkhani [1,†], Daibin Yang [2], Stephen Donovan [3] and Alan G. Taylor [1,*]

1   Cornell AgriTech, School of Integrative Plant Science, Horticulture Section, Cornell University, New York, NY 14456, USA; hsm1@cornell.edu (H.M.); ma862@cornell.edu (M.A.)
2   Institute of Plant Protection, Chinese Academy of Agricultural Sciences, Beijing 100193, China; yangdaibin@caas.cn
3   The Center for Forensic Science Research & Education, Willow Grove, PA 19090, USA; sdonovan@ptd.net
*   Correspondence: agt1@cornell.edu; Tel.: +1-315-787-2243
†   These authors contributed equally in this study.

**Abstract:** Seed treatments are frequently applied for the management of early-season pests, including seed-borne pathogens. However, to be effective against internal pathogens, the active ingredient must be able to penetrate the seed coat. Tomato seeds were the focus of this study, and the objectives were to (1) evaluate three coumarin fluorescent tracers in terms of uptake and (2) quantify seed coat permeability in relation to lipophilicity to better elucidate chemical movement in seed tissue. Uptake in seeds treated with coumarin 1, 120, and 151 was assessed by fluorescence microscopy. For quantitative studies, a series of 11 *n*-alkyl piperonyl amides with log $K_{ow}$ in the range of 0.02–5.66 were applied, and two portions, namely, the embryo, and the endosperm + seed coat, were analyzed by high-performance liquid chromatography (HPLC). Coumarin 120 with the lowest log $K_{ow}$ of 1.3 displayed greater seed uptake than coumarin 1 with a log $K_{ow}$ of 2.9. In contrast, the optimal log $K_{ow}$ for embryo uptake ranged from 2.9 to 3.3 derived from the amide series. Therefore, heterogeneous coumarin tracers were not suitable to determine optimal log $K_{ow}$ for uptake. Three tomato varieties were investigated with the amide series, and the maximum percent recovered in the embryonic tissue ranged from only 1.2% to 5%. These data suggest that the application of active ingredients as seed treatments could result in suboptimal concentrations in the embryo being efficacious.

**Keywords:** tissue lipophilicity; systemic uptake; coumarin; piperonyl amides





## 1. Introduction

Seed-borne pathogens are responsible for the initiation of numerous plant diseases and are one of the primary mechanisms for the global spread of plant pathogens [1–4]. Internal infection of seeds and colonization of the embryo and endosperm are most often associated with infection of the mother plant via the xylem, stigma, or non-vascular tissue [4–6]. Seed-borne pathogens have been observed in the seed embryo, storage tissue (endosperm and perisperm), and seed coat or testa [4,7]. Disinfection techniques can be used to remove and clean contaminants from the seed surface; however, plant pathogenic organisms located within the seed endosperm and embryo are much more difficult to control. Tomatoes are an important high-value vegetable crop and are susceptible to multiple pathogens. Tomato seeds can harbor fungal, bacterial, and viral pathogens [8–10]. Several systemic conventional pesticide seed treatments are available for fungal pathogens of tomato, but options are more limited for organic production and control of bacterial pathogens [3,11,12].

Seed treatments are applied worldwide for crop protection against pests and plant pathogens [13,14]; however, the systemic uptake and distribution of active ingredients of pesticide seed treatments in seed tissue have not been as well defined as root and leaf

transport [15,16]. The long-term efficacy of seed treatments and control of seed-borne pathogens are dependent on seed coat permeability, as the active ingredients must be able to penetrate the seed coat and diffuse to the embryo. There have been several studies focused on the physiochemical barriers that prevent or allow a chemical to permeate the seed coats of several plant species [17–19]. Taylor and Salanenka (2012) developed a system to classify seed coat permeability based on the passage of ionic and non-ionic compounds through the seed coat of ten plant species from seven plant families [18]. Tomato seeds have selective permeability defined as only non-ionic compounds diffusing through the seed coat, while ionic compounds are blocked [17,18].

Seed uptake research on potential chemical pesticides applied as seed treatments is problematic due to the potential human risk of exposure to agrochemicals and/or radioactively labeled compounds. Fluorescent tracers provide an alternative approach, and coumarins are one group that includes several fluorescent, non-ionic tracers differing in chemical properties and that allows for a more comprehensive analysis of seed coat permeability characteristics [16]. Therefore, coumarin compounds were used both for qualitative uptake [16,17,19] and, using a single coumarin compound, for quantitative uptake research [20]. One objective of this research is to use three coumarin compounds with different chemical properties for tomato seed uptake to assess optimum log $K_{ow}$.

A key chemical property that affects the uptake of an organic compound in a seed is the log $K_{ow}$, also known as the log $P$ [20–23]. A compound's lipophilicity is measured as the log $K_{ow}$ and is the ratio of its chemical concentration in octanol (o) to its concentration in the aqueous (w) phase expressed on a log10 scale [24]. A series of fluorescent piperonyl amides were synthesized, and a novel combinatorial pharmacokinetic technique was developed to provide a range of compounds with log $K_{ow}$ from 0.2 to 5.8. This series of fluorescent piperonyl amides was used to explore seed coat permeability and systemic uptake in soybean and corn seeds [23]. This same approach was adopted for tomato seed in this study.

Understanding the chemical/physical properties associated with the uptake of active ingredients in tomato seed tissue will aid in the development of new products for the control of internal seed-borne pathogens. The key objectives of this study were to evaluate the movement of selected coumarin compounds in uptake by fluorescence imaging and assess the role of log $K_{ow}$ in seed tissue permeability using a homologous series of 11 fluorescent piperonyl amides quantified by high-performance liquid chromatography (HPLC).

## 2. Materials and Methods

### 2.1. Fluorescence Microscopy of Coumarin 1, 120, and 151 in Tomato Seeds

The first study was on the uptake of selected coumarin tracers in tomato seeds imaged by fluorescence microscopy. Tomato seeds of the variety "Hypeel 696" were provided by Seminis, Oxnard, CA, and coumarin 1, 120, and 151 were purchased from TCI America, Portland, OR. The chemical and other properties of these three coumarin compounds are shown in Table 1. Tomato seeds were treated with 3 µmoles of each coumarin per gram of seed, which was 0.833, 0.631, and 0.825 mg coumarin 1, 120, or 151, respectively, per gram of seed. Each coumarin compound was mixed with 3.8 mg L650 seed treatment binder (Incotec, Salinas, Canada), 250 µL deionized water, and mixed in a 50 mL centrifuge tube using a vortex mixer (Scientific Industries, Inc., Model 2-Genie No. G560, New York, NY, USA). Ten non-treated and treated tomato seeds of each tracer were sown in 20% moisture content silica sand (#1 Q-ROK, 0.15–0.84 mm, New England Silica, Inc., South Windsor, CT, USA) and maintained in a germinator at 20 °C for 40 h in the dark. Imbibed seeds were then removed and washed with deionized water, and then the seeds were dissected with scalpel blades and imaged under an Olympus microscope (SZX12, Tokyo, Japan), imaging camera (Infinity 3- 3URC, Lumenera Corp., Ottawa, ON, Canada), and Infinity Analyze (Revision 6.5.2, Teledyne Lumenera, Ottawa, ON, Canada). Seed tissue was illuminated

with long UV light, UV lamp (Model 9-circular illuminator, Stocker & Yale, Salem, NH, USA). Non-treated seeds were used as the control.

**Table 1.** Physical/chemical properties of coumarin 120, 151, and 1.

| Coumarin Compound | CAS Number | MW, g/mol | * Log $K_{ow}$ | * Water Solubility, Log S | Excitation/Emission Max, nm | Molar Abs Coefficient, cm$^{-1}$ | Quantum Yield |
|---|---|---|---|---|---|---|---|
| 120 | 26093-31-2 | 175.2 | 1.25 | 1.25 | 342/409 | $3.50 \times 10^8$ | 0.63 |
| 151 | 53518-15-3 | 229.2 | 1.62 | −3.56 | 364/460 | $4.58 \times 10^8$ | 0.53 |
| 1 | 91-44-1 | 231.3 | 2.90 | −3.69 | 369/431 | $4.63 \times 10^8$ | 0.73 |

* Log $K_{ow}$ and water solubility data obtained from Chemicalize, ChemAxon's cheminformatic tool. Excitation/emission, molar absorbance coefficient, and quantum yield information were obtained from Aazam (2010) [25] and Taniguchi and Lindsey (2018) [26].

### 2.2. Chemicals and Synthesis of N-alkyl Piperonyl Amides

An experimental series of *n*-alkyl piperonyl amides developed by S. Donovan and B. Black [23] was used in this study. There were 11 custom synthesized homologous piperonyl amides with carbons ranging from 1 to 14, with molecular weights of 189.2 to 361.5 g/mol. The methods and materials are described in Yang et al. (2018B) [23]. Briefly, 3.0 g of piperonylic acid was added to 5 mL of thionyl chloride. The solution was then refluxed for 30 min after which 5 mL pyridine, 25 mL toluene, and 18.1 mM amines were added and the solution was refluxed for 1 h. After cooling to ambient temperature, ethyl acetate (50 mL) was added and the solution was washed with saturated NaCl, 5% NaOH, and 5% HCl. The solution was then dried (using anhydrous sodium sulfate), filtered, and concentrated using a rotary evaporator. Recrystallization was achieved by refluxing 50 mL of methylcyclohexane until a solution was attained. The C1, C2, C3, and C4 *n*-alkyl piperonyl amides were made by adding a small amount of methylene chloride until the desired solution was completed. Lastly, all solutions were cooled and vacuum filtered before use. Each piperonyl amide (0.56 mM) solution consisted of 70% acetone + 30% water.

A short octadecyl-poly (vinyl alcohol) column was used to determine the log of the octanol-water partition coefficient for each compound by HPLC [27,28]. The HPLC-log $K_{ow}$ of the *n*-alkyl piperonyl amides series is shown in relation to the number of carbon groups and water solubility determined by Chemicalize, ChemAxon's cheminformatics tool (Figure 1).

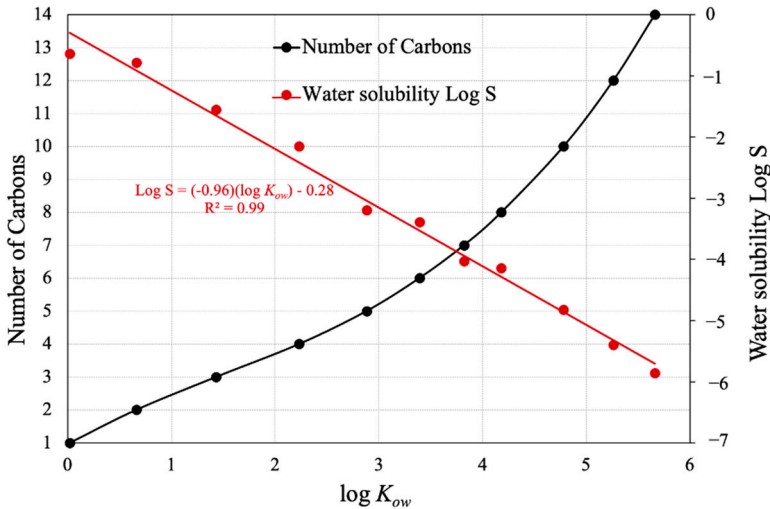

**Figure 1.** The log $K_{ow}$ of the piperonyl amide fluorescent tracers with corresponding number of carbon atoms and water solubility, log S.

*2.3. Sample Preparation for High-Performance Liquid Chromatography (HPLC) Analysis*

2.3.1. Coating Tomato Seeds with Amides

Tomato seeds "Florida 47" and "Hypeel 696" were donated by Seminis, Oxnard, CA, and "OH88119" was provided by The Ohio State University, Columbus, OH. A seed coating formulation was developed to apply high loading rates of the fluorescent tracer series as a single seed treatment. A thin adsorbent seed coating was first applied to single seeds to facilitate the high loading rates of the fluorescent tracer series in a single seed treatment. General methods and materials are described in Yang et al. (2018B) [23]. Twenty grams of diatomaceous earth (DE) was dispersed in 80 g of 4% polyvinyl alcohol (PVA) aqueous solution to prepare a 20% DE suspension concentrate. One gram of tomato seeds and 1.5 g of 20% DE suspension concentrate were stirred until a layer of dry DE was coated on the surface of each seed. The coated seeds were allowed to dry in a gentle air stream. A 1.2 mL solution of amides (approximately 6 µL for each seed) was loaded gradually onto 200 tomato seeds with a micropipette. The resulting dosage was 1 µmole of each amide per gram of seed—applied to each tomato variety. Now considering the molecular weights of the 11 amides, which ranged from 179.2 to 361.5 [23], the seed treatment dosage ranged from 0.179 to 0.361 mg per gram of seed. The seeds were again dried with a gentle air stream.

2.3.2. Incubation and Harvest of Treated Seeds in Growth Chamber

Seeds treated with the piperonyl amide series were imbibed as described in Section 2.1. Seed tissue was separated after imbibition, just prior to visible germination. Seeds were removed and washed (to remove seed treatment) with sterile distilled water, cut with a razor blade, and the embryo was removed. Embryos of 50 seeds were pooled to comprise one replicate. The endosperm and seed coat of 50 seeds were also pooled together as one sample. Three replicates were evaluated for each treatment. Ten seeds were pooled together as one sample or replicate. The covering layers consisted of the endosperm and seed coat, while the internal tissues were comprised of the embryo.

2.3.3. Harvesting Tomato Seed Tissue for HPLC Analysis

For each embryo sample, 1.5 mL of acetonitrile (MeCN) was added and the embryos were homogenized with a glass rod. For each sample containing the endosperm and seed coat, the samples were frozen with liquid nitrogen and then homogenized in a mortar after which 1.5 mL of MeCN was added [23]. The homogenized samples were vortexed for 2 min. The extract was transferred into a tube containing 20 mg of PSA, 5 mg of GCB, and 50 mg of MgSO4, then shaken for 1 min, and was then passed through a 0.22 µm syringe filter. The recovery is shown in Table S1 in the supporting information of Yang et al. (2018B) [23].

The tomato embryo and internal tissue samples were extracted by the QuEChERS (Quick, Easy, Cheap, Effective, Rugged, and Safe) method as described for soybean and corn seeds [23]. Ten tomato embryos or ten tomato endosperm + seed coats were placed into a frozen mortar and frozen with liquid nitrogen, and ground into a fine powder. The powder was transferred into a 50 mL centrifuge tube with a screw cap, and 8 mL of MeCN was added and the mixture was shaken for 2 min using a Vortex mixer at room temperature. Following this, a mixture of 2.5 g of $MgSO_4$ and 1.0 g of NaCl was added. The tube was immediately shaken vigorously for 1 min to prevent the formation of $MgSO_4$ agglomerates and centrifuged at 3500 rpm for 5 min. Then, 3.0 mL of the supernatant was subjected to dispersive solid-phase extraction (SPE) using a mixture of 8 mg GCB, 50 mg PSA, and 100 mg $MgSO_4$. The mixture was shaken vigorously for 1 min using a Vortex mixer. Finally, the extract was filtered through a 0.22 µm syringe filter. In developing the HPLC method, the percent recoveries were determined for the eleven amides from soybean embryo + testa (seed coat), and corn endosperm + embryo, and pericarp + testa. The recovery at ≤3.82 log $K_{ow}$ for both seed tissues was > 82% for soybean and > 85% for corn [23].

### 2.3.4. HPLC Analysis of Tomato Seed Tissue

The amides content was determined using an Agilent 1100 HPLC equipped with a 1200 fluorescence detector (FLD) using an ODS-3 column (GL Sciences Inc., 5 μm, 4.6 mm × 75 mm column). The wavelengths of FLD were set at 292 nm (excitation) and 340 nm (emission). The mobile phase used was 0 min 30% MeCN + 70% water, 22 min 40% MeCN + 60% water, 25 min 80% MeCN + 20% water, 40 min 90% MeCN + 10% water. The temperature of the column was 30 °C. The injected volume was 20 μL. The retention time in minutes for each amide derivative was 3.63 (C1), 5.94 (C2), 10.38 (C3), 15.43 (C4), 18.59 (C5), 21.04 (C6), 23.15 (C7), 24.96 (C8), 27.70 (C10), 31.08 (C12), and 35.58 (C14).

### 2.3.5. Tomato Seed Coat Permeability Data Calculation

Percent uptake in relation to the maximal log $K_{ow}$ (relative amount)
$$= \frac{\text{Concentration of each amide in tissue}}{\text{Concentration of the amide at the maximal log } K_{ow} \text{ in the same tissue}} \times 100\%$$
Percent uptake based on amount applied (uptake efficiency)
$$= \frac{\text{Amount of each amide absorbed by a seed}}{\text{Applied amount of each amide}} \times 100\%$$
Percent in embryo of total seed uptake
$$= \frac{\text{Amount of each amide in the embryo}}{\text{Sum amount of each amide in the covering + internal tissues}} \times 100\%$$

## 3. Results

### 3.1. Fluorescence Microscopy of Coumarin 1, 120, and 151 in Tomato Seeds

Assessment of coumarin 1, 120, and 151 uptake was conducted by visual fluorescence observation of the applied seed treatment tracers in tomato seed tissue. Coumarin tracers evaluated in this study were all non-ionic and therefore were expected to permeate the seed coat and move to the embryo [17]. Results showed that coumarin 120, with the greatest water solubility and the lowest log $K_{ow}$ (Table 1), was readily taken up in the embryonic tissue, whereas coumarin 1 and coumarin 151 were only partially taken up in the embryo (Figure 2). The low level of fluorescence in the non-treated control was attributed to autofluorescence in the embryonic tissues.

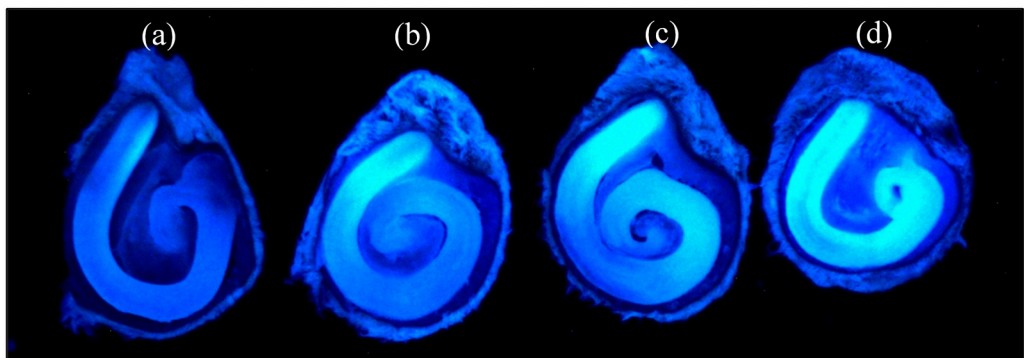

**Figure 2.** Tomato "Hypeel 696" seed coat permeability of three different coumarin tracers: (**a**) non-treated, (**b**) coumarin 1, (**c**) coumarin 151, and (**d**) coumarin 120.

### 3.2. Tomato Seed Coat Permeability

#### 3.2.1. Maximal Uptake of Piperonyl Amides in Relation to log $K_{ow}$

The maximum (100%) relative amount of *n*-alkyl piperonyl amide recovered in tomato seed tissue was in the range of 2.88–3.39 log $K_{ow}$ in embryonic seed tissue and 3.39–4.18 log $K_{ow}$ in endosperm + seed coat tissue (Figures 3 and 4). Amide diffusion to the embryo was limited when log $K_{ow}$ exceeded 4.18 (Figure 3). The maximal uptake in relation to log $K_{ow}$ was achieved at lower log $K_{ow}$ values for Hypeel 696 in both types of seed tissue compared with OH88119 and Florida 47, which had very similar uptake profiles (Figures 3 and 4).

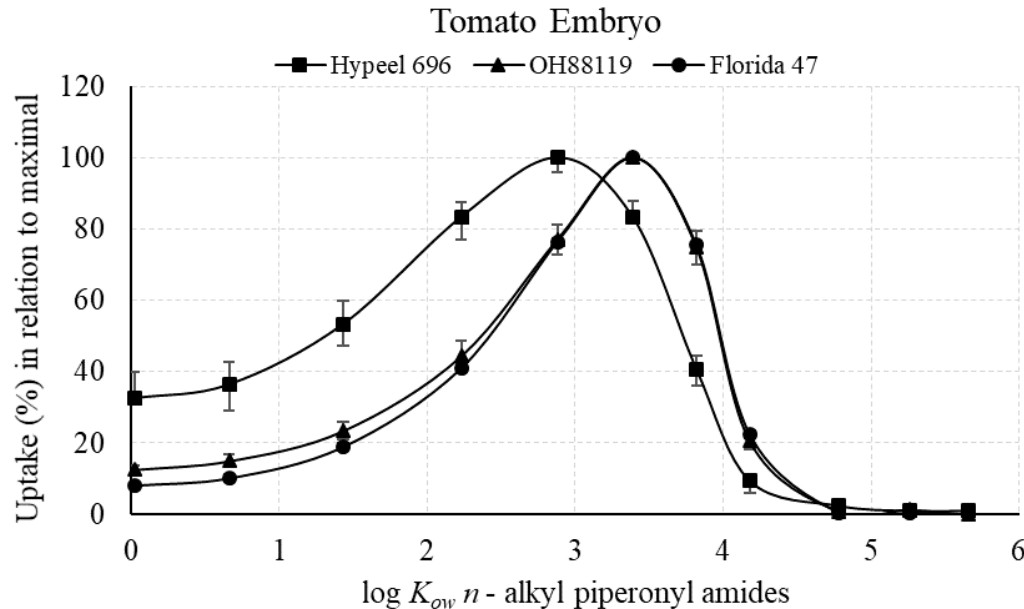

**Figure 3.** *N*-alkyl piperonyl amide uptake in tomato embryo in relation to maximal log $K_{ow}$ of 100%. Means with standard error bars are shown.

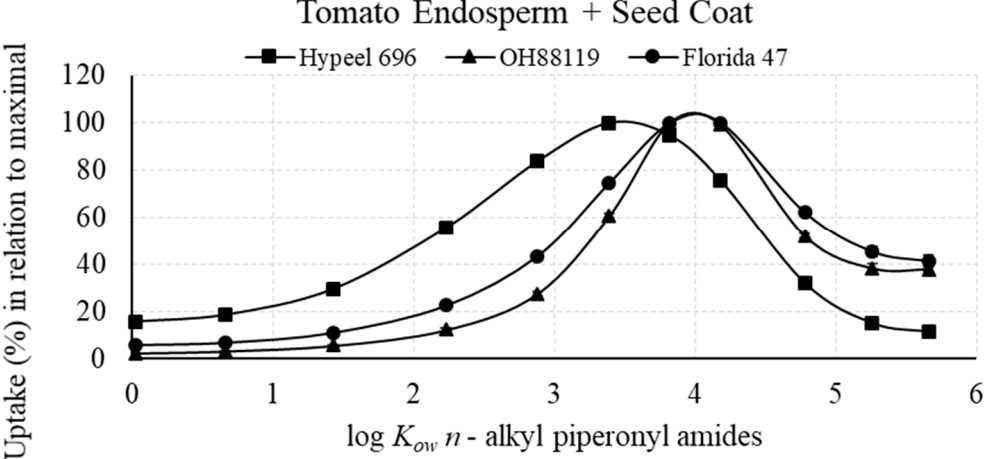

**Figure 4.** *N*-alkyl piperonyl amide uptake of tomato seed coat + endosperm in relation to maximal log $K_{ow}$ of 100%. Means with standard error bars are shown.

### 3.2.2. Uptake Efficiency (%) of Piperonyl Amides in Seed Tissue in Relation to Amount Applied

The uptake efficiency, based on the total amount recovered in the embryo compared with the amount applied, showed that maximum uptake associated with log $K_{ow}$ for the embryo occurred at 2.88 for Hypeel 696 and 3.39 for OH99119 and Florida 47 (Figure 5). However, even at the maximal log $K_{ow}$, the percent uptake was only 5.0% for Florida 47, 4.3% for Hypeel 696, and 1.2% for OH99119. In contrast, uptake efficiency for the entire seed was much greater than the embryo, and ranged from 27% to 36% for the three varieties (Figure 6).

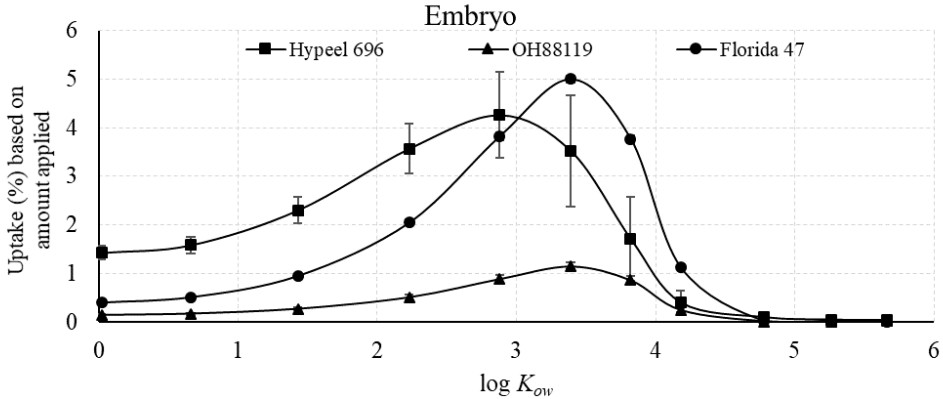

**Figure 5.** Uptake efficiency of piperonyl amides in the embryo, measured as percent compound applied of *n*-alkyl piperonyl amides. Means with standard error bars are shown.

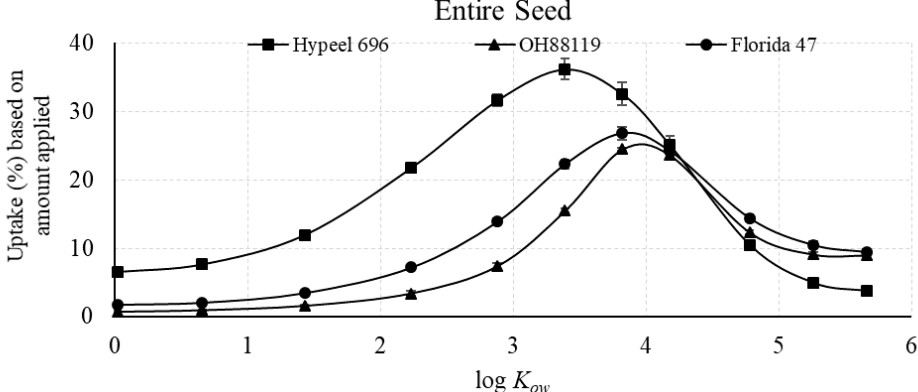

**Figure 6.** Uptake efficiency of piperonyl amides in the seed, measured as percent compound applied of *n*-alkyl piperonyl amides. Means with standard error bars are shown.

3.2.3. Percent of Piperonyl Amides in the Embryo Compared with the Entire Seed

The percent of the lipophilic amide series in the tomato embryo declined with log $K_{ow}$ from 0.02 to 4.18 for Hypeel 696 and OH8819 (Figure 7). In contrast, Florida 47 revealed a slight increase in the percent embryo distribution from log $K_{ow}$ 0.02 to 2.88–3.18, followed by a decrease to 4.78.

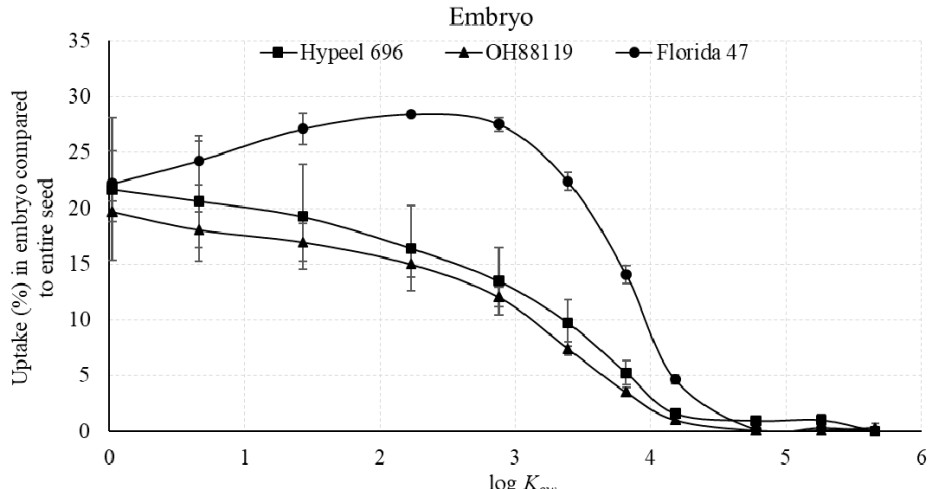

**Figure 7.** Percent of the absorbed *n*-alkyl piperonyl amides in seed embryo compared with the entire seed uptake of three tomato varieties. Means with standard error bars are shown.

## 4. Discussion

Fluorescent tracers were used in many previous studies in our lab to examine seed coat permeability in vegetable and field crop seed species. Application of single tracer compounds was used in these qualitative studies to determine seed coat permeability characteristics, resulting in three categories: (1) permeable, (2) selectively permeable, and (3) non-permeable [17]. A dual fluorescent tracer method was later developed to investigate corn pericarp/testa permeability of 27 maize lines [19]. This method could be readily adopted to determine the seed coat permeability category of other seed species. Collectively, both tomato [17] and corn [19] have selective permeability as only non-ionic compounds diffused through the seed coat, while ionic compounds were blocked. A single coumarin compound, coumarin 120, was used in quantitative uptake studies, and a linear increase in seed uptake was measured for corn seed treatment dosage in the range of 0.01 to 1.0 mg coumarin 120 applied per gram of seed [20]. In this study, coumarin 120 and each amide in the series were applied at 3 and 1 μmole per gram of seed, respectively. These seed treatment dosages convert to a range of 0.83 to 0.17 mg per gram of seed, which was in the linear uptake range of corn [20].

The objective of the first investigation in this study was the evaluation of the uptake of three coumarin fluorescent tracers in an attempt to develop a simple method to assess the optimum log $K_{ow}$ for the penetration of neutral compounds through the tomato seed coats. The major advantage was the use of readily available chemical compounds, and these tracers were previously documented with systemic uptake in seeds and seedlings [16]. In addition, fluorescence microscopy could be used for rapid assessment for comparisons without the need for chemical extraction and chemical analyses. Unfortunately, fluorescence intensity was more related to water solubility than log $K_{ow}$ (Table 1 and Figure 2). Moreover, limited conclusions can be drawn using only three tracer compounds. These inconclusive results were attributed to the use of heterogeneous compounds with different physical/chemical properties (Table 1). In addition, there are other properties unique to each coumarin compound including polarizability, topological polar surface area (TPSA), polar surface area (PSA), calculated molar refractivity (CMR), the number of hydrogen bond donors and acceptors, and pK$_a$ (Chemicalize, ChemAxon's cheminformatics tool), and these properties may play a role in seed uptake. Moreover, fluorescence microscopy images may produce false-positive images with the confounding effect of auto-fluorescence from internal seed structure constituents.

There is great value in understanding the chemical/physical properties of active ingredients, and this information can guide a directed chemical synthesis program giving optimal uptake. Alternatively, potential uptake of an existing active ingredient with known or predicted log $K_{ow}$ can be assessed through knowledge of the optimal lipophilicity for seed coat permeability. For this second objective, a combinatorial pharmacodynamic technique was employed using a homogeneous series of 11 *n*-alkyl piperonyl amides that varied in log $K_{ow}$ from 0.02 to 5.66. The mixture of amides was applied as a seed treatment, and tomato seeds were imbibed and dissected into two portions, the embryo, and the endosperm + seed coat. The relative amounts of the amides in these two fractions were quantified by HPLC and plotted as a function of log $K_{ow}$. This allowed a clear understanding of the role of lipophilicity as it relates to uptake through the tomato seed coat and endosperm and the resulting transport into the embryo by neutral compounds. This knowledge of the optimal physical properties is an invaluable guide in the targeted control of internal seed-borne pathogens.

The overall uptake profile for the tomato embryo and endosperm + seed coat of all three varieties revealed a Gaussian distribution (Figure 3) that is similar to root uptake in plants [21–23]. This similar Gaussian distribution for both tomato roots and seeds was expected based on the composition of the barrier layers. Suberin is found in the endodermis and exodermis of tomato roots [29] and also the inner layer of the tomato seed coat [30]. In contrast, the Gaussian distribution pattern in root uptake in corn and soybean was not revealed for corn or soybean seed uptake [23].

The maximal uptake for the tomato embryo tissue of the three varieties ranged from 2.88 to 3.39 log $K_{ow}$ (Figure 3), while the maximal uptake ranged from 3.39 to 3.88 log $K_{ow}$ for the endosperm + seed coat (Figure 4). Thus, a slight shift to lower log $K_{ow}$ for the embryo tissue in comparison with the other seed tissues was revealed. In the case of corn, the maximal uptake was 3.39 log $K_{ow}$ for both the endosperm + embryo and pericarp/testa using a similar method with the 11 *n*-alkyl piperonyl amides [20]. Therefore, both tomato and corn have similar maximal uptake profiles. Unfortunately, the tomato embryo readily detached from the endosperm during dissection of the fully imbibed seed, which did not allow the measurement of the sum of endosperm with embryo, so a comparison of the effect of the endosperm on shifting the maximal log $K_{ow}$ could not be directly made between corn and tomato.

The uptake efficiency was calculated as the percent of each amide taken up in relation to the amount applied. The maximum uptake efficiency of the entire seed of the three varieties ranged from 27% to 36% (Figure 6), while the maximum uptake efficiency of corn was 43% [20]. Therefore, tomato had lower seed coat permeability than corn, which may be attributed to seed coat composition. The inner layer of the tomato seed coat is known as the semipermeable layer [31] and was shown to be composed of suberin [30], while corn has a semipermeable cutinized or suberized membrane that is located below the inner integument [32].

The maximum uptake efficiency of the embryo in relation to the amount applied of the three varieties ranged from 1.2% to 5.0% (Figure 5). Another calculation based only on the absorbed *n*-alkyl piperonyl amides uptake revealed that less than 30% of an amide was measured in the embryo compared with the entire seed (Figure 7). These data demonstrate that most of the piperonyl compounds were unable to reach the embryo. However, fluorescence images revealed the greatest fluorescence intensity in the embryo compared with the endosperm or seed coat (Figure 2). Therefore, fluorescence imaging that provides excellent qualitative data on the presence or absence of a tracer in the seed tissue was not related to quantitative results from our analytical method.

The pathway by which the applied seed treatment moved to the embryo was not investigated in this study. In the dicot seed *Sedum acre*, movement between the seed compartments was attributed to symplastic movement with cell-to-cell movement through the plasmodesmata [33]. We assume that movement from the endosperm to embryo in tomato seed is by the same symplastic pathway.

Three tomato varieties were investigated with the 11 *n*-alkyl piperonyl amides. The maximal log $K_{ow}$ for both the embryo and endosperm + seed was shifted to a slightly lower value for Hypeel 696 compared with the two other varieties (Figures 3 and 4). Florida 47 had the greatest accumulation in the embryo with 5.0% (Figure 5), while Hypeel 696 had the greatest accumulation in the entire seed with 36% (Figure 6). After an amide was absorbed, Florida 47 had a greater distribution in the embryo than the other two varieties (Figure 7). These varietal differences may be attributed to differences in seed coat composition and/or structural properties. Varietal differences in tomato seed coat permeability were related to the efficacy of jasmonic acid seed treatments used as an elicitor of defense against western flower thrips [34]. The thickness and compactness of the inner tomato seed coat layer composed of suberin [28] may be responsible for varietal differences. Further, the composition of the seed coat and embryo may differ and thus can affect both permeability and affinity for a compound. Further study is needed to investigate varietal differences in seed uptake, as retention of an active ingredient in the seed coat could result in a suboptimal concentration in the embryo being efficacious.

## 5. Conclusions

This study quantitively described the relationship between the log $K_{ow}$ and the permeation capacity of a chemical through the seed coat to the embryo of tomato seeds. The relatively hard and thick tomato testa attenuated the movement of the seed treatment to the embryo tissue. Less than 5% of the applied compound was measured in the embryo,

while most resided in the seed coat + endosperm. For the control of internal seed-borne pathogens, seed treatment with log $K_{ow}$ in the range of 2.9 to 3.8 log $K_{ow}$ is suggested as these chemicals were found to most effectively reach the tomato embryo tissues.

The piperonyl amide method uses a combinatorial pharmacodynamic technique to probe the uptake and transport of xenobiotic compounds in seeds. This is in contrast with the use of heterogeneous compounds that differ in a multitude of physical properties, isolation efficiencies, and detection sensitivities. When using heterogeneous compounds, often the experimental method involves a separate experiment for each compound to generate an uptake and/or transport parameter. Combining the ensemble of data from the piperonyl amide method resulted in a trend that identified the optimum chemical properties for uptake and accumulation in specific seed tissues. Moreover, the combinatorial pharmacodynamic method used eleven piperonyl amides combined into a single experiment, with significant benefits with regard to time, cost, and many experimental variables being eliminated. Further, very subtle absorption and transport trends were quantified in different crop seeds [23], and also in plants and insects [Donovan and Black, unpublished]. Thus, the method can be broadly adapted for agricultural research, and provides detailed physical property space information at a level of precision that is not available using other techniques.

**Author Contributions:** Conceptualization, A.G.T. and S.D.; methodology, H.M., M.A. and D.Y.; software, H.M. and M.A.; validation, A.G.T., H.M. and M.A.; formal analysis, H.M. and M.A.; investigation, H.M., M.A. and D.Y.; resources, A.G.T.; data curation, A.G.T., S.D., H.M., M.A. and D.Y.; writing—original draft preparation, H.M. and M.A.; writing—review and editing, A.G.T., S.D., H.M., M.A. and D.Y.; visualization, A.G.T. and M.A.; supervision, A.G.T.; project administration, A.G.T.; funding acquisition, A.G.T. All authors have read and agreed to the published version of the manuscript.

**Funding:** This work was supported by the Specialty Crop Research Initiative (grant no. 2015-51181-24312) from the USDA National Institute of Food and Agriculture. D. Yang was partially supported by the China Scholarship Council (grant No. 201503250009). This material is based upon work that is supported by the National Institute of Food and Agriculture, US Department of Agriculture, Multi-state Project W-4168, under accession #1007938.

**Institutional Review Board Statement:** Not applicable.

**Informed Consent Statement:** Not applicable.

**Data Availability Statement:** Not applicable.

**Acknowledgments:** Authors would like to thank Lailiang Cheng, Cornell University, for providing access to his HPLC for this project.

**Conflicts of Interest:** The authors declare no conflict of interest.

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
