# Peer review of "Tomato Seed Coat Permeability: Optimal Seed Treatment Chemical Properties for Targeting the Embryo with Implications for Internal Seed-Borne Pathogen Control"

_agriculture, doi:10.3390/agriculture11030199_

Round 1

Reviewer 1 Report

Review score of the paper agriculture-1128807 titled “Tomato Seed Coat Permeability and Optimal Seed Treatment Chemical Properties for Targeting Internal Seed Borne-Pathogens” and authored by Mayton, et al.

The paper is an original research about extremely relevant topic of early season pest management, including seed-borne pathogens, that are affecting tomato worldwide. More precisely, the paper aimed to evaluate three coumarin fluorescent tracers on the uptake and quantification of seed coat permeability in relation to lipophilicity to better elucidate chemical movement in seed tissues.

The manuscript is well organized and written, However, the language used in the manuscript is not up to the standard, and several major and minor mistakes are shown, and many of the phrases are not clear. It should go through a complete proof-reading by a native speaker.

In the manuscript, as far as my understanding goes, some drawbacks and shortcuts are present and they must be faced (and solved).

The other major concern is the formatting of manuscript according to author’s guidelines. There are several mistakes regarding references sequence in the text as well as in the citations section.

I have some other major suggestions as follows

However, I'm not convinced that the title, as it is, fully reflects the content of the manuscript

Line 17: Uptake in seeds treated with coumarin 1, 120, and 151 were assessed. Units???

Keywords must be different from title words

Say something about the importance of tomato crop at the start of introduction section, which is in fact the ultimate part of this study.

The introduction section needs some serious revisions. There are some grammatical errors which make the sentences very difficult to understand. There are some outdated and irrelevant literature as well in this section. Correlate your objectives of the study with the future prospects of this experiment at the end of introduction section.

The paper is generally well organized and written, but discussion of results is extremely long and it is very engaging for the reader to remain focused on. This is one case when the Results and Discussions sections could be merged together in order to better show and discuss the many (interesting) topics one by one.

References are not properly formatted regarding journal abbreviated names etc.

Author Response

Response to reviewer #1: Authors of this manuscript appreciate the positive feedback from Reviewer #1 for a detailed reading of our manuscript and for the valuable comments. Our response (in red) follows:
Point 1: The other major concern is the formatting of the manuscript according to the author’s guidelines. There are several mistakes regarding reference sequence in the text as well as in the citations section.
Response 1: The article reformatted and the authors used the templated of Agriculture Journal. 
Point 2: However, I'm not convinced that the title, as it is, fully reflects the content of the manuscript
Response 2: Title changed. Line 2-3: Tomato Seed Coat Permeability: Optimal Seed Treatment Chemical Properties for Targeting the Embryo with Implications for Internal Seed-Borne Pathogen Control
Point 3: Line 17: Uptake in seeds treated with coumarin 1, 120, and 151 were assessed. Units???
Response 3: Line 19 In this part of the study we assessed the uptake of each tracer for seeds of different tomato varieties under microscope visually, and it’s not a quantitative measurement so there is no unit.
Point 4: Keywords must be different from title words
Response 4: Line 29: Changes have been made. “seed” and “tomato” deleted from the keywords. And the title edited “Tomato Seed Coat Permeability: Optimal Seed Treatment Chemical Properties for Targeting the Embryo with Implications for Internal Seed-Borne Pathogen Control”
Point 5: Say something about the importance of tomato crop at the start of introduction section, which is in fact the ultimate part of this study.
Response 5: Introduction edited please see lines 32-44 and 79-81.
Point 6: The introduction section needs some serious revisions. There are some grammatical errors which make the sentences very difficult to understand. There are some outdated and irrelevant literature as well in this section. Correlate your objectives of the study with the future prospects of this experiment at the end of introduction section.
Response 6: Authors rewrote the introduction from lines 32-44 and 79-81. Grammatical errors edited throughout the manuscript such as line 19 “was used instead of were” and line 83 “a used instead of an”
Point 7: The paper is generally well organized and written, but the discussion of results is extremely long and it is very engaging for the reader to remain focused on. This is one case when the Results and Discussions sections could be merged together in order to better show and discuss the many (interesting) topics one by one.
Response 7: Authors followed the Agriculture journal’s guidelines and preferred to keep the results and discussion separated.  
Point 8: References are not properly formatted regarding journal abbreviated names etc.
Response 8: All references (Page 12-13) edited and properly formatted.

Reviewer 2 Report

The manuscript "Tomato Seed Coat Permeability and Optimal Seed Treatment Chemical Properties for Targeting Internal Seed Borne-Pathogens" is of interest. The originality of the study and the novelty it brings in the field is of actuality. The paper is well structured, the abstract is concise and in the topic; the introduction is supported by well selected bibliographic data. The Experimental and Modeling Approach correctly. Results and Discussions could be improved by studying other papers in the field.

I suggest to include information regarding advantages, limitation of the method used, time, costs, etc., this will increase the manuscript value.

The paper is not identify clearly implications for research, practice and society. I recommend to state clearly the conclusions of the study to enriched your work.

Please use the template of the journal.

Author Response

Response to reviewer #2: The author's thanks Reviewer #2 for thoughtful comments and constructive suggestions on this manuscript.

The manuscript "Tomato Seed Coat Permeability and Optimal Seed Treatment Chemical Properties for Targeting Internal Seed Borne-Pathogens" is of interest. The originality of the study and the novelty it brings in the field is of actuality. The paper is well structured, the abstract is concise and in the topic; the introduction is supported by well-selected bibliographic data. The Experimental and Modeling Approach correctly. Results and Discussions could be improved by studying other papers in the field.

Point 1: I suggest including information regarding advantages, limitation of the method used, time, costs, etc., this will increase the manuscript value.

Response 1: According to point 1 of reviewer #2 Line 573-581 edited. The advantage is the piperonyl amide method can quantitively describe the relationship between the log P and the permeation capacity of a chemical. Regarding the time and cost savings advantage of the combinatorial pharmacodynamic method using piperonyl amides method that we used in this study, additional explanation added to line 585-594.

Point 2: The paper does not identify clearly implications for research, practice, and society. I recommend stating clearly the conclusions of the study to enriched your work.

Response 2: The conclusion section was revised to describe the implications of this study. Please see lines 573-594.